# Avoidance of non-localizable obstacles in echolocating bats: A robotic model

**Carl Bou Mansour**[1], **Elijah Koreman**[2], **Jan Steckel**[3], **Herbert Peremans**[4],
**Dieter Vanderelst**[5]*

**1** Department of Psychology, University of Cincinnati, Cincinnati, Ohio, United States of America,
**2** Department of Computer Science, Cornell University, Ithaca, New York, United States of America,
**3** Constrained Systems Lab, University of Antwerp, Antwerp, Belgium, **4** Department of Engineering
Management, University of Antwerp, Antwerp, Belgium, **5** Department of Biological Sciences, University of
Cincinnati, Cincinnati, Ohio, United States of America

* vanderdt@ucmail.uc.edu

## Abstract

Most objects and vegetation making up the habitats of echolocating bats return a multitude
of overlapping echoes. Recent evidence suggests that the limited temporal and spatial reso-
lution of bio-sonar prevents bats from separately perceiving the objects giving rise to these
overlapping echoes. Therefore, bats often operate under conditions where their ability to
localize obstacles is severely limited. Nevertheless, bats excel at avoiding complex obsta-
cles. In this paper, we present a robotic model of bat obstacle avoidance using interaural
level differences and distance to the nearest obstacle as the minimal set of cues. In contrast
to previous robotic models of bats, the current robot does not attempt to localize obstacles.
We evaluate two obstacle avoidance strategies. First, the Fixed Head Strategy keeps the
acoustic gaze direction aligned with the direction of flight. Second, the Delayed Linear Adap-
tive Law (DLAL) Strategy uses acoustic gaze scanning, as observed in hunting bats. Acous-
tic gaze scanning has been suggested to aid the bat in hunting for prey. Here, we evaluate
its adaptive value for obstacle avoidance when obstacles can not be localized. The robot's
obstacle avoidance performance is assessed in two environments mimicking (highly clut-
tered) experimental setups commonly used in behavioral experiments: a rectangular arena
containing multiple complex cylindrical reflecting surfaces and a corridor lined with complex
reflecting surfaces. The results indicate that distance to the nearest object and interaural
level differences allows steering the robot clear of obstacles in environments that return
non-localizable echoes. Furthermore, we found that using acoustic gaze scanning reduced
performance, suggesting that gaze scanning might not be beneficial under conditions
where the animal has limited access to angular information, which is in line with behavioral
evidence.

## Author summary

The sonar system of bats provides only limited information about the location of obsta-
cles. In particular, it is unlikely that bats can localize multiple, complex obstacles that

**Data Availability Statement:** All data and
computer code are available at https://zenodo.org/
record/3551169#.XfJCJfxOmUk (DOI:10.5281/
zenodo.3551169).

**Funding:** The authors received no specific funding for this work.

**Competing interests:** The authors have declared that no competing interests exist.

return a multitude of interfering echoes. Nevertheless, these animals can fly swiftly through densely cluttered habits. To explain how bats do this, we proposed they only need to compare the loudness of the echoes at the left and the right ear. If the echoes at the left ear are louder than at the right ear, obstacles are probably located to the left. Therefore, the bat should bank right (and vice versa). In this paper, we test whether such a simple strategy would allow bats to avoid obstacles. We equipped a robot with a sonar system resembling that of a bat, and we implemented the obstacle avoidance strategy above. We tested this robotic bat in environments mimicking those used in experimental studies of their sonar behavior. The robot was able to avoid most of the obstacles in both environments. Therefore, we conclude that bats could rely on a simple strategy when avoiding obstacles in complex environments.

## Introduction

Echolocating bats rely on their biosonar systems to avoid obstacles in complex environments [1]. In previous work [2], we suggested that a strategy based on the interaural level difference of the onset of the echo train can support obstacle avoidance. We proposed that the bat could compare the intensity of the echo onset in the left and the right ear. If the onset of the echo train is louder in the left (right) ear, the bat turns right (left). The turn magnitude was determined by the distance to the nearest obstacle, as conferred by the first echo. Our computer simulations suggested that such straightforward phonotaxis can steer a bat clear from obstacles in large simulated environments. This simple strategy does not require the bat to reconstruct the 3D layout of the obstacles from the echoes, which is a notoriously hard problem (See [2] for arguments and references), especially in highly cluttered environments where obstacle avoidance is paramount. Hence, the strategy is robust as it can be used to guide flight under conditions where little or no angular information can be extracted from echoes. It is also computationally undemanding, which might reduce reaction time and make it compatible with the requirement to respond quickly to obstacles while negotiating cluttered environments.

In our previous simulations [2], we assumed that the bat's gaze and flight direction always coincide. However, several studies on prey capture behavior have documented that a bat's acoustic gaze often deviates from its flight direction (for example, [3–5]). Ghose et al. [6] and Falk et al. [7] found that, while hunting for prey, the bat's flight direction is determined by its gaze direction through a Delayed Linear Adaptive Law (DLAL). The bat's flight direction was found to follow its gaze direction with a delay. Interestingly, this pattern of gaze direction leading action has also been documented concerning eye movements in humans (See [8] for a review).

More recently, we have used simulations to propose a sensorimotor model of the prey capture by echolocating bats [9]. The simulations in that paper did incorporate the DLAL. The simulated bat was assumed to steer its gaze to keep the prey in the center of its field of view. As in the experiments reported by Ghose et al. [6] and Falk et al. [7], the flight direction followed the gaze direction. To assess the contribution of the DLAL to prey capture, we also ran simulations in which the head and body were rigidly coupled (i.e., as in our obstacle avoidance paper, [2]). Based on these results, we concluded that coupling the flight direction and gaze direction through a DLAL allows bats to keep erratically moving prey in their field of view, and thereby, increases the probability of successful prey capture. This previous work hinted at a clear functional advantage of a loose coupling between flight and gaze direction through a DLAL.

In the current work, we extend our previous model of obstacle avoidance in bats to include the DLAL. This extension allows evaluating whether this steering mechanism is compatible with obstacle avoidance. In particular, we investigate whether relaxing the coupling between gaze and flight direction has a functional advantage when avoiding obstacles in densely cluttered and demanding conditions.

In contrast to our previous computer simulations, the current paper employs a robotic model. Robotic models have been used before to test hypotheses about animal behavior, for example [10, 11]. Compared to computational models, robotic models are especially helpful when modeling the physics and dynamics of the animal's interaction with the environment is difficult, e.g., [12]. In this case, computational models often have to resort to simplifications, which may limit the validity of the results (See [13] for examples).

Veracious modeling of the propagation and reflection of bat echolocation signals is computationally demanding, especially for the complex environments we set out to study in this paper. Using a physical model, we do not have to introduce simplifying assumptions about the physical interaction of the sonar signals with the environment. In contrast, the echoes returning from the environment capture the full complexity of the echoes faced by bats.

In our earlier work [2], we simulated large artificially generated environments consisting of point reflectors. Here, we test the obstacle avoidance strategy in two environments that mimic highly cluttered setups commonly used in behavioral experiments. First, we evaluate our obstacle avoidance strategy in a rectangular arena, modeling a flight cage with densely packed obstacles. The number of obstacles has been chosen to reflect conditions that induced a significant collision rate in hunting bats [7]. Second, we evaluate the obstacle avoidance strategies in a narrow corridor lined with reflectors, for example, [14–16]. Importantly, recent neurophysiological [14] and behavioral evidence [15, 17] confirms that the echoes from the closely spaced reflectors lining the corridor are not separable by the bat. Instead, as stated in [14], responses to echo cascades from the densely spaced reflectors represent a single extended stimulus event that lasts over 40 ms. Similarly, the stochastic reflectors, i.e., egg cartons, lining our test corridor, provide an environment that generates many highly overlapping echoes that cannot be individually localized. Hence, this environment presents a good test case for an obstacle avoidance strategy not relying on obstacle localization.

In summary, the goals of this paper are threefold. First, we aim at establishing that the interaural level difference based steering, hitherto only tested in simulation, can be implemented in a physical system using real echoes. Second, this paper tests interaural level difference based steering in highly cluttered environments shown behaviourally and neurophysiologically to be demanding and returning non-localizable echoes. Third, we wish to evaluate whether acoustic gaze steering, observed in hunting bats, is compatible with interaural level difference based obstacle avoidance.

## Materials and methods

### Robotic platform

The Amigobot (Adapt Mobilerobots) was used as a robotic platform (Fig 1). This is a differential drive robot measuring approximately $33 \times 28 \times 15$ cm. The robot was equipped with an onboard single board computer running Ubuntu Linux handling data acquisition and controlling the robot, through a serial interface. In addition, we mounted a custom-built sonar data acquisition board (DAQ) on the robot. The DAQ featured one DAC channel (sample frequency 360 kHz) and two ADC channels (sample rate 300 kHz). The DAC channel was used to drive an ultrasonic transducer (MA40S4S Murato), producing narrowband ultrasonic signals centered around 40 kHz.

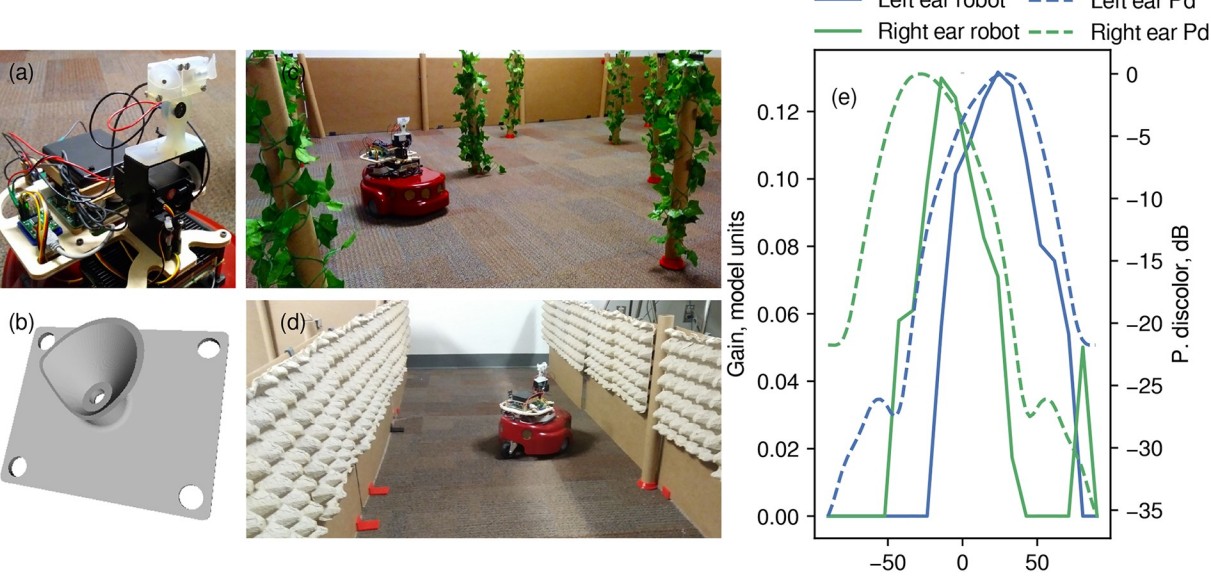

**Fig 1.** (a) Close up of the robot showing the housing containing the emitter and two microphones in artificial pinnae. Below this, the pan-tilt system can be seen. (b) 3D rendering of one of the artificial pinnae. (c) View of the robot in the rectangular arena. (d) View of the robot in the corridor lined with egg cartons. (e) The gain of the ears as a function of azimuth angle. This plot depicts the output of the cochlear model for a single cardboard tube reflector at 1 m distance from the robot showing the directionality of the robot's sonar system. Overlaid is the directionality of the bat *P. discolor* at elevation = 0° and 40 kHz, simulated data from Vanderelst et al. [18].

The emitter used in this work has a smaller diameter ($\sim$ 10 mm) than the broadband emitters typically used on robotic models of bat echolocation, e.g., [19–23]. Therefore, the width of its emitted beam is more comparable to that of bats. Fitting a piston model to the beam emitted by various species of bats, Jakobsen et al. [24] found pistons with a diameter between about 5 and 10 mm to best fit the data. The obvious downside of this emitter is its limited bandwidth. However, our previous work [2] suggests that, at least for obstacle avoidance in the horizontal plane, broadband signals are not necessary. Indeed, the obstacle avoidance strategy we proposed previously does not exploit spectral cues.

The emission and the echoes were recorded using two Knowles (FG—23742-D36) microphones. These were embedded in 3D printed bat-like stylized pinnae (Fig 1). The artificial pinnae were printed on a Form 2 (Formlabs) printer at a resolution of 50 microns. As illustrated by comparison with the directionality of the bat *P. discolor* [18], the stylized pinnae gave the microphones a batlike HRTF with a clear IID gradient across the frontal hemisphere (Fig 1). The housing containing the emitter and microphones was mounted on a pan-tilt system driven by two servo motors. This allowed us to simulate the head movements observed in bats, e.g., [3, 4, 16].

## Emissions and echo processing

In cluttered environments, bats typically increase their emission rate, in part by emitting calls in so-called strobe groups, for example, [25, 26]. Short intragroup intervals characterize Strobe-groups. For example, Sandig et al. [26] report on the change in call interval as bats approach an obstacle array. In their data, bats approaching a challenging array of obstacles increased their call rate from about 20 Hz (about 2 m from the obstacles) to about 100 Hz (right before passing the obstacles). Similar call rates have been reported by other authors, e.g., [7, 27–29].

In our robotic model, the call rate sets both the rate of data acquisition and processing thereof. It is unlikely that bats can process echoes quickly enough to update their motor behavior every 10 ms. From experiments, it seems that bats require between 50 (to abort a hunting sequence, [30]) and 20 (to exhibit a reflex, [31]) ms to respond to echoes. Hence, we modeled a fixed interpulse interval of 50 ms.

Bats change the duration of their calls as a function of the environment or the stage of the prey capture sequence (for example, *Eptesicus fuscus*, [32]). However, our robotic model operating in fixed high-clutter environments, we used a constant emission duration. We excited the emitter for 0.3 ms. Due to the response characteristics of the emitter, the resulting emission lasted about 1.5 ms, which is in the range of call durations used by *Eptesicus fuscus* under lab conditions, e.g., [32].

After each emission, the signals recorded at both microphones were passed through a single channel cochlear model [33] consisting of a gammatone filter with a center frequency of 40 kHz followed by half-wave rectification. The result was compressed (exponent 0.4). Finally, a low-pass filter with a 1kHz cut-off was applied to the signals.

To detect the echoes in the output of the cochlear model, we used a peak finding method to find local maxima in the signal. The first peak to surpass a threshold in either ear was considered as the arrival time of the first echo. The energy in the left and the right envelope was integrated across 300 samples (1 ms), starting 150 samples before the arrival time of the first echo (i.e., the peak of the first local maximum in the signal). This yielded an estimate of the loudness of the first echo in the left and the right ear. The arrival time of the first echo for the $i$th call yielded an estimate of the distance $\hat{d}_i$ to the closest obstacle. The processing of the echoes is illustrated in Fig 2.

## Rectangular arena

The robot's ability to avoid obstacles was tested in a $3 \times 4$ m rectangular arena bounded by 50 cm high corrugated cardboard panels. The obstacles consisted of 50 cm long cardboard tubes with a diameter of 4 cm (Fig 1). This mimicked experimental conditions under which bat echolocation behavior has been studied before. For example, Falk et al. [7] evaluated the echolocation behavior of *E. fuscus* in a flight cage populated with 20 cm diameter artificial trees made from mist net wrapped around two metallic rings creating a cylinder that hanging from the ceiling. Petrites et al. [25], Barchi et al. [29] and Knowles et al. [15] used 4 cm wide chains as obstacles (See [2, 34] for more early references). However, the tubes being simple human-

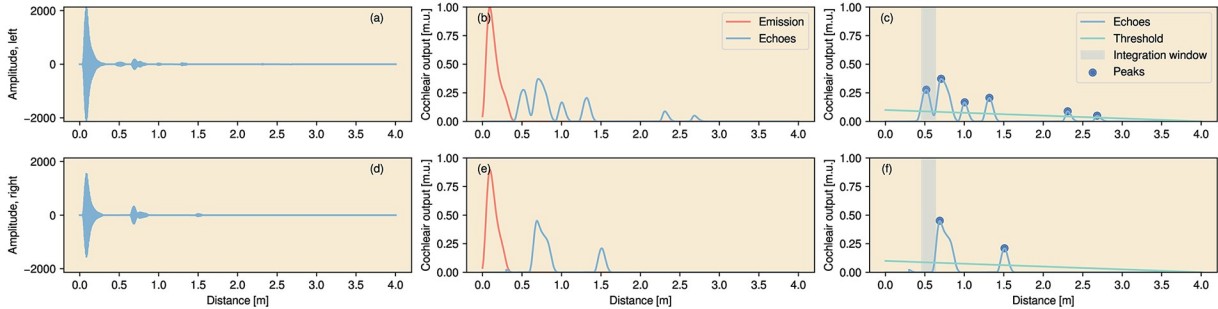

**Fig 2. Illustration of the echo processing.** (a) The signal at the left ear, containing the emission and the echoes. (b) The output of the cochlear model [33]. The emission (shown in red) is subtracted from this result. (c) Peaks in the output of the cochlear model that surpass a given threshold (in green) are considered as echoes. The energy in a window of 1 ms around the arrival time of the first echo in the left or the right ear is integrated. (d-f) Same for the right ear. In this example, the first detectable echo arrives at the left ear.

made obstacles do not return the multiple echoes that are characteristic for vegetation (compare, [14, 35, 36]). Therefore, we wrapped the cardboard tubes in artificial ivy vines (Fig 1). The obstacles were pseudo-randomly distributed through space.

We used 11 cardboard tubes (i.e., $\approx 1/m^2$) spaced less than a 1 meter apart to create a challenging environment for our robot. For example, Falk et al. [7] tested bats in an arena with a lower density of obstacles (12 obstacles; arena area: 36 $m^2$; density: 0.3/$m^2$; Minimum spacing: 1 m). While hunting for prey in this environment, bats crashed in 20% of the trials (each of which lasted typically 10 seconds, personal communication, Falk. 2019). Barchi et al. [29] flew bats in arenas with a similar density to ours (18 chains; arena area: 18 $m^2$; density: 1/$m^2$). However, in their experiments, the obstacles were not uniformly distributed in space. The clustering of the obstacles resulted in greater distances between them than in our experiment. The trajectory of the robot in the arena, the boundaries, and location of the obstacles were digitized using a Vive Tracker (HTC) mounted on top of the robot.

## Corridor

In addition to the rectangular arena, we also tested the robot's ability to follow a corridor lined with obstacles. Mimicking the experiments of, for example, Knowles et al. [15], who studied the echolocation behavior of *Eptesicus fuscus* in complex cluttered corridors, we constructed a corridor about 90 cm wide and 3 m long. The walls of the corridors consisted of the same corrugated cardboard panels as used in the arena described above. In addition, we lined the walls with egg cartons (Fig 1). The structure of the egg cartons, forming complex reflectors, resulted in the walls returning multiple (overlapping) echoes (See Fig 3).

## Control strategies

**The DLAL obstacle avoidance strategy.** Bats lower their flight speeds in cluttered environments [7, 25, 26], and experience with a particular environment seems to increase the flight speed [15]. In flight cage experiments introducing varying levels of clutter, the flight speed of

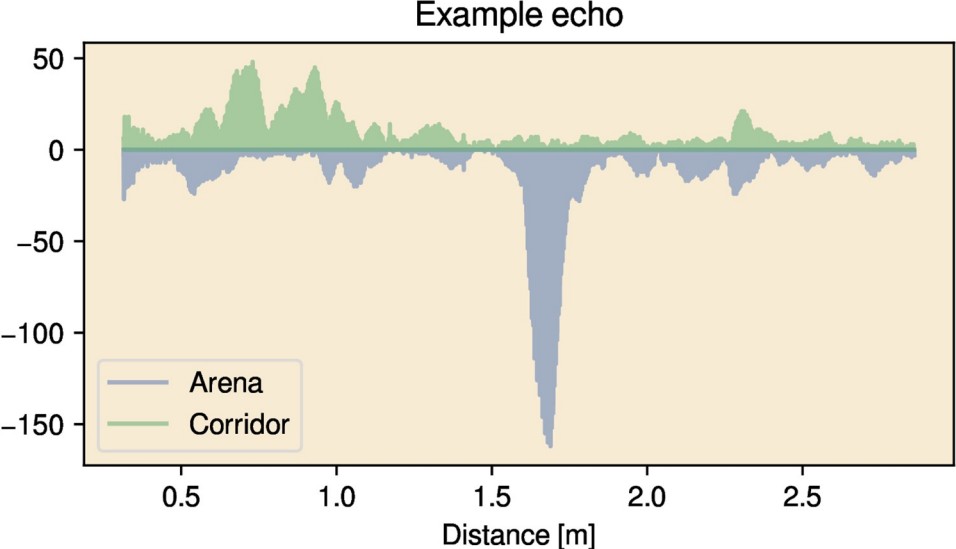

**Fig 3. One example of an echo train collected in each environment (rectangular arena and corridor).** These examples illustrate that the environments return multiple (overlapping) echoes for every call. The emission has been omitted from these plots to make the echoes more visible.

*E. fuscus* has been reported to vary from about 2.5 ms$^{-1}$ [7, 29] to 5 ms$^{-1}$ [25]. However, when executing sharp turns, bats' speed can be lower than 1 ms$^{-1}$, even for bats with a relatively high body mass such as *E. fuscus*, which are typically fast fliers [15, 37].

In our experiments, which feature a very demanding environment, we aimed to obtain an average speed of about 2 ms$^{-1}$, i.e., the lowest speed at which *E. fuscus* would fly continuously in wind tunnel experiments [38]. The speed also corresponds to the lower range of the flight speed of *E. fuscus* recorded in the experiments of Knowles et al. [15] and it is slightly below the flight speed of the bats in the experiments of Falk et al. [7] (2.49 ms$^{-1}$) and Knowles et al. [15] (2-m ms$^{-1}$).

We further modulated the speed of the robot based on the gaze angle $\phi$, i.e., the angle between the drive direction of the robot $\theta$ and the emission direction. The rationale for this is that, when the gaze angle is large, the bat needs to make a sharp turn, and it will slow down [39, 40]. We simulated a maximum speed of 3 ms$^{-1}$ for $\phi = 0°$ and 1 ms$^{-1}$ at $\phi = 45°$ (see Fig 4a).

After each call, the head of the robotic bat was turned to a new orientation $\phi_i$ with respect to the body. The angular rotation $\Delta\phi_i$ of the sonar head was determined by the distance $\hat{d}_i$ and the relative loudness in both ears. If the echo was loudest in the left ear, the head turned to the right ($\Delta\phi_i < 0$). If the right ear received the loudest echo, the head turned to the left ($\Delta\phi_i > 0$). The magnitude of the rotation $\Delta\phi_i$ was determined by the distance $\hat{d}_i$.

Closer echoes resulted in a larger rotational speed of the head. To obtain this, the magnitude of change in head orientation $\Delta\phi_i$ between pulses varied from 50° to 25° (Fig 4c). Fig 4d shows that $\dot{\phi}_i$ varied from about 1000°/*s* to 500°/*s*.

The simulated range for the angular velocity of the head $\dot{\phi}_i$ (Fig 4d) matches the available behavioral data. Seibert et al. [3] measured the differences in gaze angle as *Pipistrellus pipistrellus* flew rapidly through different outdoor environments at an average speed of over 5ms$^{-1}$. The largest difference in gaze angle between subsequent calls was about 40°. As the bats emitted calls at a rate slightly over 10 Hz, this implies that the bats rotated their head at maximum speeds of about 400°/*s*. On the other hand, Sumiya et al. [41] report on the gaze shifts of *Pipistrellus abramus* while hunting for prey. Their data implies angular rotation velocities up to about 3000°/*s*. Therefore, the range for the angular velocity $\dot{\phi}_i$ used in this paper seems attainable by bats. The maximum angle of the head $\phi_i$ was limited to ± 90 degrees.

Ghose et al. [6] and Falk et al. [7] found that the flight path of hunting bats is linked to its gaze direction by a DLAL. In line with the model they propose to capture this behavior, we modeled the rotation of the robot $\Delta\theta_i$ after call *i*,

$$\Delta\theta_i = k \cdot \phi_\tau \tag{1}$$

In Eq 1, *k* is a gain parameter. The term $\phi_\tau$ denotes the orientation of the head $\phi$ (with respect to the body) at time $\tau$ before the current call *i*. The parameters *k* and $\tau$ that best describe the behavior of the bat have been found to vary across conditions and stage of prey capture [6, 7]. In this paper, we fix $k = 10$ and $\tau = 50$ ms. These values correspond to the highest *k* and the lowest $\tau$ reported and imply a tight coupling between head and body [6, 7]. The value $\phi_\tau$ was obtained by interpolating the past values for the orientation of the head $\phi$.

**Fixed Head Strategy.** The obstacle avoidance strategy described in the previous paragraphs was compared with a Fixed Head Strategy. Under this strategy, the gaze and the flight direction were always aligned (i.e., the strategy we reported in ref. [2]). Including this strategy allows us (1) to confirm that the results obtained in simulation can be replicated using real echoes and in more densely cluttered environments, (2) evaluate the contribution of the DLAL to obstacle avoidance.

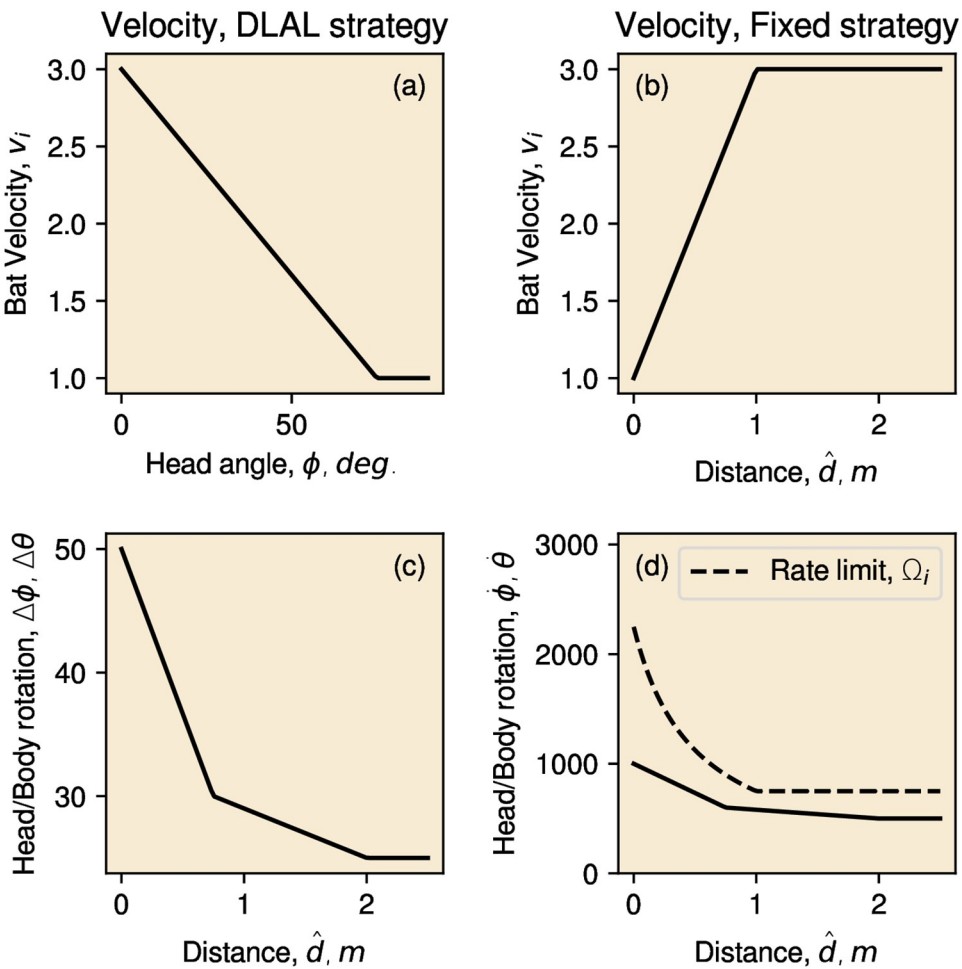

**Fig 4. Curves depicting the relationship between the variables in our models.** (a) The simulated velocity for the DLAL Strategy as a function of distance $\hat{d}_i$. (b) The simulated velocity for the Fixed Head Strategy as a function of distance $\hat{d}_i$. (c) The head rotation applied between calls for the DLAL Strategy as a function of the closest echo distance $\hat{d}_i$. The body rotation applied for the Fixed Head Strategy as a function of the closest echo distance $\hat{d}_i$. (d) The change in the head (DLAL Strategy) or body (Fixed Head Strategy) orientation between calls (panel a) together with the interpulse interval modeled, implies a rotational velocity of the head (DLAL Strategy) or body (Fixed Head Strategy). This panel plots the implied velocities. This panel also shows the maximum body rotational velocity as observed across bat species flying at different speeds by Holderied et al. [42]. In generating this curve, the velocity as a function of distance $\hat{d}_i$ for the Fixed Head Strategy is used (panel b).

Using the Fixed Head Strategy, the rotation of the body $\Delta\theta_i$ was a function of the distance to the nearest echo as depicted in Fig 4c. This imposed an increase in the rotational velocity $\dot{\theta}_i$ with decreasing $\hat{d}_i$ (see Fig 4d).

As the gaze $\phi_i$ was always zero under the Fixed Head Strategy, the flight speed was controlled based on the distance $\hat{d}_i$. Closer obstacles resulted in a reduction of the flight speed, as shown in Fig 4b.

The flight speed of bats imposes a limitation on the curvature of their flight paths [39, 40]. Holderied et al. [42] reports the maximum curvature of the flight paths of various species. These data indicate that bats' rotational velocities typically result in g-forces below 4. Giuggioli et al. [43] employed a value of 4 in their model of bat interaction. To ascertain whether the rotational velocities of the body employed in implementing the Fixed Head Strategy (Fig 4c)

were attainable by bats, we calculated the maximum rotation velocity as given by Holderied et al. [42]. Following Holderied et al. [42], the maximum rotation velocity of the body $\Omega_i$ as a function of velocity $v_i$ at call $i$ is given by,

$$\Omega_i = \frac{G \cdot 9.81}{v_i} \tag{2}$$

with $G = 4$. Because, under the the Fixed Head Strategy, velocity $v_i$ is determined by the distance $\hat{d}_i$ (Fig 4b), we can express $\Omega_i$ as a function distance $\hat{d}_i$. This curve given Eq 2 is plotted in Fig 4d, allowing it to be compared with the rotational velocity $\dot{\theta}_i$. This shows that the rotational velocities of the bat body used in implementing the Fixed Head Strategy fell within the range observed by Holderied et al. [42].

**Random Walk.** We included a random walk condition to establish a baseline for the number of collisions that can be expected in our environment without an obstacle avoidance strategy in place. This condition is an empirical approach to determining the a priori collision probability for a given environment, which is typically approximated analytically in behavioral experiments, e.g., [1, 44]. Under this condition, the simulated speed and interpulse interval (IPI) were fixed at $2\text{ms}^{-1}$ and 50 ms, respectively. During each IPI, the robot rotated over an angle sampled uniformly from the interval [25˚, 50˚], i.e., the range of body rotations used under the DLAL Strategy and Fixed Head Strategy.

## Collision detection

The width of our robotic platform is about the same as the average wingspan of *E. fuscus*. For example, Petrites et al. [25] report 30 cm as the maximum wingspan while our robot measured $28 \times 33$ cm. Therefore, we used the robot's built-in collision detection. Whenever the robot hits an obstacle (a pole or wall), the robot produces a warning sound. If this happens, we would pause the experiment (stop the robot) and record a collision. We would then turn the robot away from the obstacle and resume the experimental run. In the corridor environment, we terminated the trial upon collision. If the robot turned around and started making its way back to the beginning of the corridor, the trial was also terminated.

## Results

### Rectangular arena

We ran ten trials of about 600 calls each for the Fixed Head Strategy, the DLAL Strategy, and the Random Walk. This corresponds to about 30 seconds of simulated flight ($\delta t_i = 50$ ms) for each trial. The robot was initiated at a different randomly chosen position and orientation for each trial.

**Linear and rotational velocity.** Fig 5 depicts the descriptive statistics concerning the paths of the robot in the rectangular arena for the Fixed Head Strategy, the DLAL Strategy and the Random Walk. The average velocity was about 2 $\text{ms}^{-1}$ across strategies (Fig 5a). There was a small but statistically significant difference in velocity between the two non-random strategies (DLAL Strategy: 1.92 $\text{ms}^{-1}$, Fixed Head Strategy: 2.07 $\text{ms}^{-1}$, Kruskal-Wallis H-test, $H = 755$, $p < 0.01$). The variation in velocity was larger in the DLAL Strategy (Levene test, $W = 3186$, $p < 0.01$). This indicates that in the DLAL Strategy, the robot was somewhat slower, and its speed varied more.

The median angular velocity of the robot was about the same in both strategies (DLAL Strategy: 347˚/$s$, Fixed Head Strategy: 341˚/$s$, Kruskal-Wallis H-test, $H = 0$, $p = 0.51$, Fig 5b).

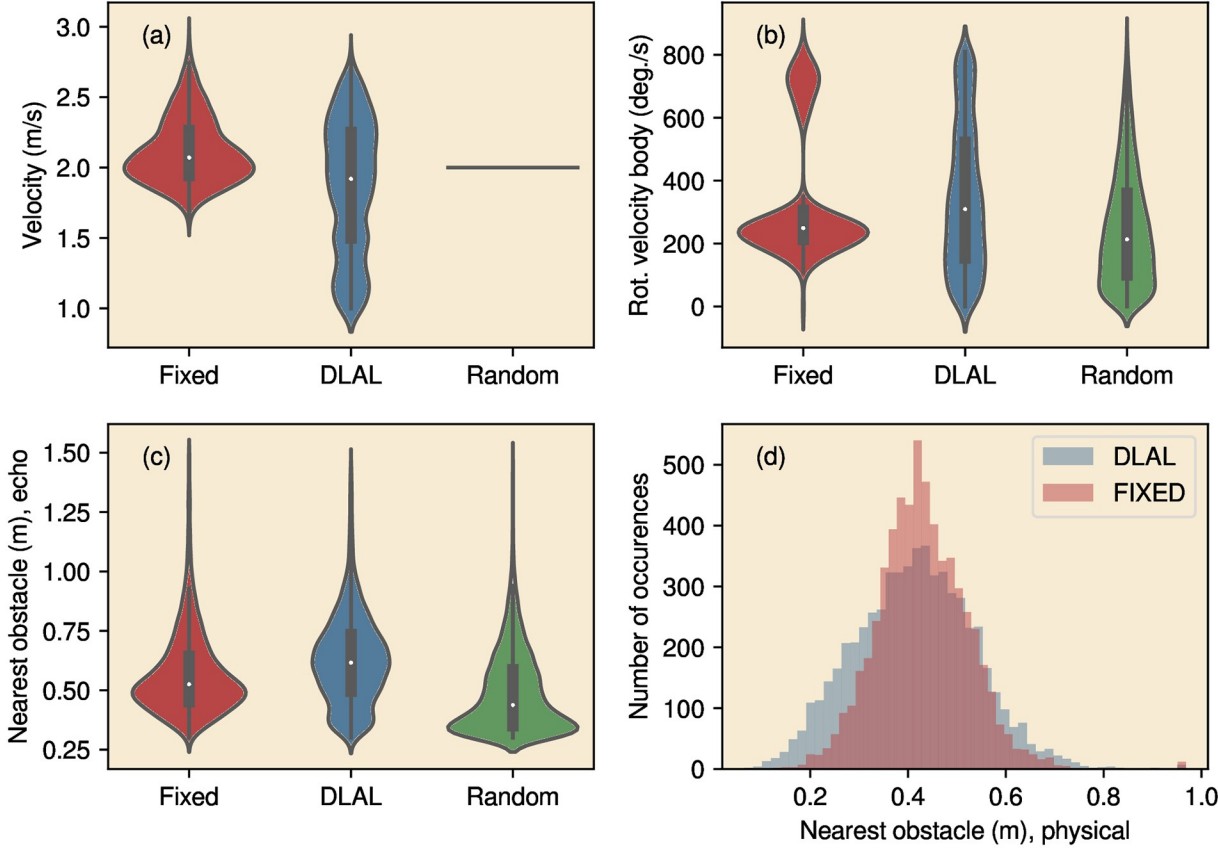

**Fig 5. Descriptive statistics of the behavior of the robot in the arena, for the three strategies (Fixed Head Strategy, DLAL Strategy and Random Walk).** (a) Violin plots of the velocity of the robot, (b) Violin plots of the rotational velocity, (c) violin plots of the distance to the nearest obstacle (as determined by the sonar), (d) distribution of the distance to the nearest obstacle (as determined by the Vive tracking system).

However, as can be seen in the graph, the Fixed Head Strategy, unlike the DLAL Strategy, resulted in a bimodal distribution of the rotational velocity of the body.

**Obstacle distance.** The distance to the nearest reflector, as determined by the sonar readings, was typically considerably smaller than 1 m (Fig 5c). The median detected distance $\hat{d}_i$ was smaller for the Fixed Head Strategy than for the DLAL Strategy (DLAL Strategy: 0.62 m, Fixed Head Strategy: 0.53 m, Kruskal-Wallis H-test, $H = 442$, $p < 0.01$). This is partly because the DLAL Strategy causes the head to be rotated away from obstacles, thereby increasing the average detected reflector distance. Inspecting the physical distances between the robot and the closest obstacle, as observed using the tracking system, we found that these were similar across conditions (medians. DLAL Strategy: 0.42 m, Fixed Head Strategy: 0.43 m, Kruskal-Wallis H-test, $H = 59$, $p < 0.01$). However, the distribution of physical distances for the DLAL Strategy showed a longer tail for smaller values (Fig 5d). Both the distances as detected by the sonar system and the distances as given by the tracking system confirmed that the robot operated in a very cluttered environment: in about 50% of the time, the closest obstacle was less than 50 cm away.

**Obstacle avoidance.** Fig 6 depicts the relative obstacle avoidance performance of the various strategies, expressed as collisions per second and as collisions per meter. Compared to the Random Walk, the DLAL Strategy resulted in an >80% reduction in collisions. The

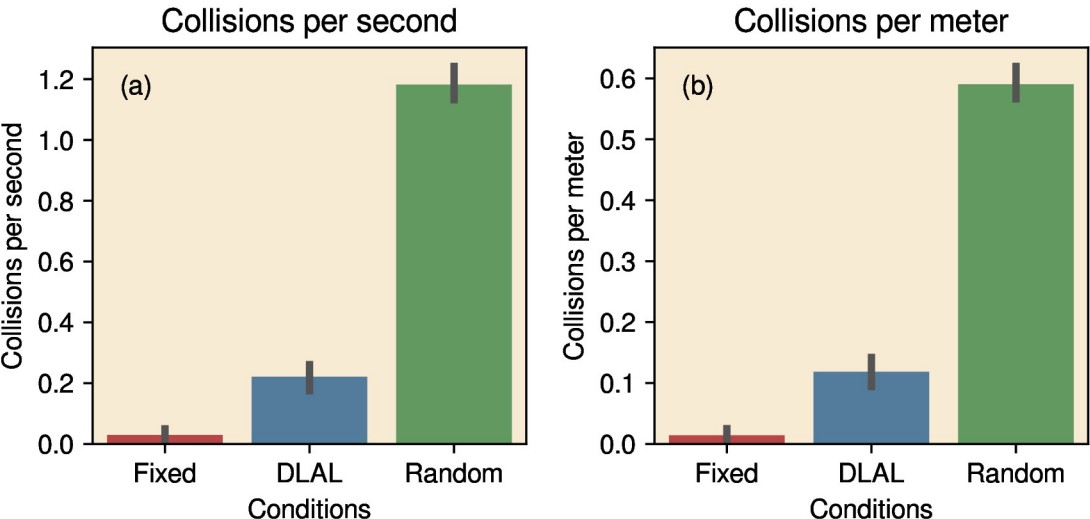

**Fig 6. The number of collisions in the arena, for each of the three different strategies.** Performance in expressed either as the number of collisions per second (a) or the number of collisions per meter driven by the robot (b). Lines indicate the 95% confidence intervals.

Fixed Head Strategy resulted in a further reduction of >80% in the number of collisions. Hence, both the Fixed Head Strategy and the DLAL Strategy significantly reduced the number of collisions. However, the Fixed Head Strategy outperformed the DLAL Strategy (97% reduction of collisions compared to the Random Walk, Kruskal-Wallis H-test, $H = 14$, $p < 0.01$).

**Trajectories.** Fig 7a and 7b shows two example trajectories of the robot through the rectangular arena using the DLAL Strategy and the Fixed Head Strategy. Qualitatively, it can be seen that both strategies lead to different behavior. The Fixed Head Strategy steers well away from obstacles. In contrast, the DLAL Strategy steers the robot much closer to obstacles.

Fig 7c also reveals that both strategies fail in different locations. The DLAL Strategy collided with obstacles at positions throughout the arena. The Fixed Head Strategy failed in corners. Right corners are challenging for an algorithm that uses interaural differences for steering. The left and the right edge of the corner form a corridor. The interaural difference based Fixed Head Strategy leads the robot down this corridor, eventually colliding where the walls meet. In

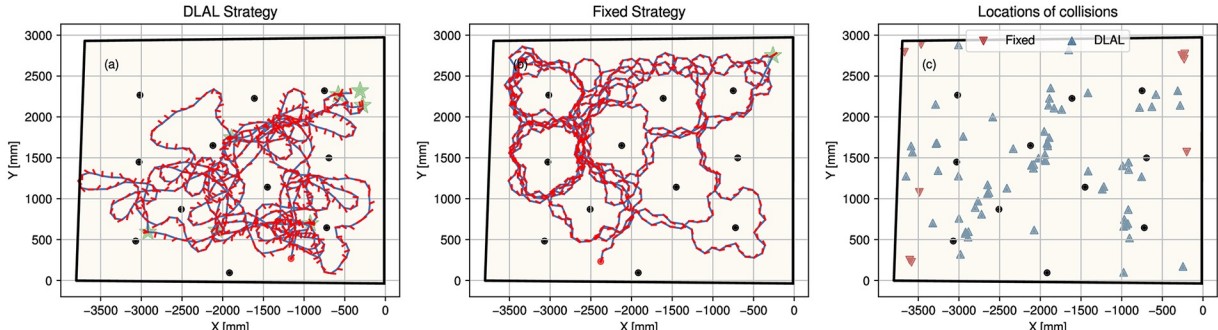

**Fig 7.** Example trajectories for the robot controlled by (a) the DLAL Strategy and (b) the Fixed Head Strategy. The arena and obstacles are depicted in black. The trajectory of the robot is drawn as a blue line. The head orientation of the robot is shown as short red lines. A green star indicates collisions with obstacles. (c) Locations of the collisions recorded for both strategies across all trials.

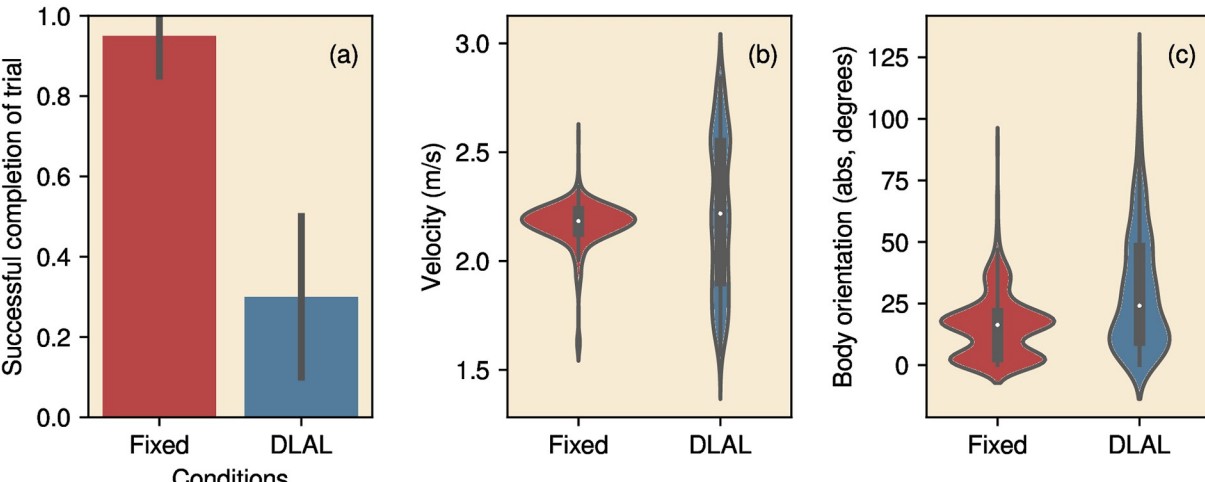

**Fig 8. Performance of the robot in the corridor environment.** (a) Proportion of trials in which the robot successfully reached the end of the corridor for the Fixed Head Strategy and DLAL Strategy. Lines indicate the 95% confidence intervals. (b, c) violin plots of the velocity and the body orientation for both strategies with respect to the corridor (0˚ is aligned with the corridor).

open-ended corridors, this behavior allows the robot to follow the corridor (see next section). However, in the case of corners, the behavior is maladaptive. The DLAL Strategy seemed to be better at avoiding being led down this trap. It avoided the corners.

## Corridor

We ran 20 trials for both the Fixed Head Strategy and the DLAL Strategy in the corridor. Fig 8 depicts the results obtained in the corridor. The Fixed Head Strategy always steered the robot successfully to the end of the corridor. In contrast, the DLAL Strategy failed to do so in 70 percent of the trials (Chi-square test for proportions, $\chi^2 = 18$, $p < 0.01$). Both strategies resulted in about the same velocity (DLAL Strategy: 2.22 ms$^{-1}$, Fixed Head Strategy: 2.17 ms$^{-1}$, Kruskal-Wallis H-test, $H = 5$, $p = 0.03$). However, the DLAL Strategy resulted in a larger variance in speed (Levene test, $W = 706$, $p < 0.01$). The trajectories of the Fixed Head Strategy were straighter than the trajectories obtained using the DLAL Strategy. This can be seen in the distribution of body orientation (Fig 8c). On average, the angle between the driving direction of the robot and the axis of the corridor was larger when using the DLAL Strategy (DLAL Strategy: 30˚, Fixed Head Strategy: 16˚, Kruskal-Wallis H-test, $H = 86.38$, $p < 0.01$).

## Discussion

### Avoidance of non-localized obstacles

In contrast to the visual system, the echolocation system has no direct access to spatial information. Angular information needs to be computed from spectral and temporal cues [45]. While bats can locate the origin of single reflectors with high precision, inferring the azimuth and elevation of a reflector is not a trivial computation [46]. Moreover, when the bat receives a cascade of overlapping echoes with limited signal-to-noise ratio, localization might be not feasible at all [2, 17, 36, 46]. Therefore, bats might be assumed to often operate under conditions where they have limited access to angular information. For example, vegetation returns a statistical ensemble of echoes rather than individual and separated echoes [35, 36, 47]. Both recent neurophysiological [14] and behavioral [15] evidence from bats flying through corridors lined with obstacles support the view that, when multiple echoes return within a window

of about 2 ms, bats are not capable of perceiving the echoes as separate reflectors. Moreover, to the best of our knowledge, no biologically plausible mechanism has been proposed that can infer the spatial layout of vegetation (or other complex reflectors) from the cascade of echoes it returns. Finally, Geberl et al. [17] recently presented evidence suggesting that the spatial resolution of bat sonar ($\sim 80°$ in azimuth) is much worse than previously assumed.

The arena and corridor used here mimic standard experimental setups and represent highly cluttered circumstances with inter-obstacle distances that are considerably smaller than in our previous simulation study [36]. Using a robot model, as opposed to numerical simulations, allowed us to model the echoes in dense environments veraciously. Moreover, just as natural reflectors [35, 36], both our arena (poles wrapped in ivy) and the corridor (walls covered with egg cartons) return a multitude of echoes with every call (see Fig 3).

Previous work on robot models of bat obstacle behavior has assumed that the bat can (approximately) localize obstacles. Both Yamada et al. [48] and Eliakim et al. [49] presented robot-based models in which obstacle location is used as input to an obstacle avoidance algorithm. Both studies estimated the azimuth location of obstacles using interaural time differences, a cue that might not be available to bats to localize individual reflectors [50, 51].

In contrast to the work of Yamada et al. [48] and Eliakim et al. [49], our approach does not assume that obstacles can be localized. Therefore it is of particular interest that the environments in which Yamada et al. [48] evaluated their robot were similar to the rectangular arena used in this paper. They used poles with a diameter of 12 centimeters, and, in some trials, the robot also had to avoid the walls circumscribing the experimental arena. The spacing between the poles was about 1 m. Our reflectors, which were wrapped in artificial ivy leaves resulted in complex echoes (Figs 3 and 2). These can be expected to result in less accurate interaural time differences, and therefore, less accurate azimuth estimates.

The maximum speed of their robot (driving 4-30 cm between calls) was higher than in the current study (driving 5-15 cm between calls). However, they do no report the average speed across trials. Despite the similarity between the arena of Yamada et al. [48] and our arenas, our robot completed the obstacle avoidance task without a need to localize obstacles. Unfortunately, it is difficult to compare the performance of both systems as Yamada et al. [48] seem to have used different numbers of poles across trials (but less than 11) scattered across different areas (but larger than our arena). Nevertheless, as a rough comparison, they note that the average duration driven without collision was 20 seconds (for their system that did not use beam scanning). In our arenas, the average time between collisions was about 30 seconds for the Fixed Head Strategy.

The results presented in this paper confirm that the robust interaural difference based obstacle avoidance strategy, previously proposed in simulation [2], steers the robot away from obstacles, even under very demanding conditions. This simple strategy manages to avoid over 90% of the obstacles that would be hit by driving randomly through the arena. The algorithm also guided the robot through a corridor lined with stochastic reflectors. These results confirm that an interaural difference based obstacle avoidance strategy is robust under circumstances where obstacle localization is jeopardized, which are the circumstances under which a robust obstacle avoidance strategy would be most beneficial. Further work could aim at exploring additional robust cues for obstacle avoidance. Even simpler, monaural strategies can be imagined to be part of this stack (See [52] for an example).

## The Delayed Linear Adaptive Law (DLAL) Strategy and Fixed Head Strategy

Gaze scanning in bats has been likened to saccadic eye movements in mammals [3, 4, 53]. In humans, eye scan paths seem mostly determined by top-down control in function of the

current task demands (See [8] for an example). In various tasks, eye scan paths have been shown to lead action. For example, objects are fixated before they are picked up. Moreover, scan paths are different when participants are asked to answer different questions about a scene. In addition to this top-down control, limited effects of low-level image saliency have been observed [8, 54].

Irrespective of whether top-down or bottom-up control steers eye movements, planning saccades requires the availability of angular information. If the visual system would have no access to the azimuth/elevation location of visual features, task-relevant objects or salient regions, it could not plan a path to the region of interest. In the visual system, angular information is directly available through the layout of the sensor surface (retina) of the eye.

In the echolocation system, both interaural level differences and interaural time differences provide angular information. Given the small size of the bat's head and time-intensity trading occurring in the neural responses to received echoes, it has been argued that the most robust source of angular information available to bats is the interaural level difference [51]. On the other hand, when confronted with complex echo signals, bats possess neural populations that code for interaural time differences of the envelopes of these complex echoes [50]. The acoustic attention scheme proposed by Simmons et al. [55] describes how time and intensity cues can be consistently combined, whereby level differences amplify, through time-intensity trading, the physically occurring time differences. However, in our simple controller, implementing the same time-intensity trading mechanism would result in the smaller interaural time difference cues being dominated by the interaural level difference cues. Hence, as both cues are highly correlated, we have chosen to make use only of the relative loudness of echoes in the left and the right ear as a robust (although, not perfect) indicator about whether the reflectors are more likely to be located left or right from the midline. Furthermore, this cue would be available even under very cluttered conditions. Therefore, the current implementation of the DLAL Strategy can be seen as a model of gaze direction control in echolocating bats requiring minimal (but robust) spatial information.

While using the interaural differences to guide the gaze might be the only option available to bats when clutter degrades spatial information, our results suggest gaze scanning might not be a good option. We found the Fixed Head Strategy outperformed the DLAL Strategy. This indicates that, if the complexity of the environment prevents the bat from inferring the spatial layout of the environment, gaze scanning is disadvantageous. Indeed, the limited spatial information provided by the interaural differences might not be sufficient to guide the gaze to informative directions. In particular, under these conditions, the cost of not looking where you are going might outweigh the limited benefit of looking around.

Behavioral evidence seems to support the hypothesis that, as angular information degrades, the benefit of active beam scanning decreases. Indeed, the idea that bats' gaze direction strategy might depend on the complexity of the environment, and therefore on their ability to infer its spatial layout, was also proposed by Knowles et al. [15]. Flying bats in corridors of chains, they found that the bats did not exhibit the gaze scanning observed in other experiments, e.g., [3–5]. Knowles et al. [15] hypothesized that the bats did not perceive the dense array of chains as individual objects but as a 'clutter field' (See Warnecke et al. [14] for neurophysiological evidence supporting this). The data of Knowles et al. [15] indicates that under conditions where the bats do not (or are not able to) reconstruct the spatial layout of the environment, they fix their gaze in the flight direction and abandon gaze scanning.

Our experiments included a proxy of the setup used by Knowles et al. [15]. We ran the robot through a 90 cm wide corridor, either using the Fixed Head Strategy or DLAL Strategy. The corridor was lined with textured reflectors that return multiple overlapping echoes. We found that the Fixed Head Strategy managed to follow the corridor to end consistently. In

contrast, the DLAL Strategy mostly failed either by colliding with the walls or by turning around in the corridor and driving back to the start. In addition, the Fixed Head Strategy was also found to result in a somewhat faster and more straight motion.

The behavioral data reported by Knowles et al. [15] and our results indicate that the Fixed Head Strategy is a robust approach to obstacle avoidance. We conclude that the DLAL is compatible with a simple obstacle avoidance algorithm that uses minimal spatial information to direct the gaze. However, the advantage of the Fixed Head Strategy leads us to conclude that gaze movements might reduce obstacle avoidance performance in highly cluttered environments. This could explain why Knowles et al. [15] did not observe gaze scanning behavior in their experiments.

When bats can extract angular information, this could be used to support a beam scanning strategy benefiting obstacle avoidance. Indeed, this was demonstrated by Yamada et al. [48] who, in addition to a system with a fixed gaze, also equipped their robot with a sonar system that allowed scanning. In their experiments, aiming the beam at nearby obstacles did incur a benefit and resulted in better obstacle avoidance. The average time between collisions increased from about 30 seconds to 42 seconds, probably because aiming the beam at nearby obstacles allowed these to be localized more accurately. These results confirm that, under conditions that allow the bat to localize obstacles, it could use active beam aiming to its benefit. It should be noted that the algorithm implemented by [48] was not the DLAL Strategy: the robot aimed its beam at the obstacle but turned away from them—this behavior is not possible under the DLAL Strategy as proposed by Ghose et al. [6] and Falk et al. [7].

## Conclusion

In this paper, we examined obstacle avoidance based on minimal but robust cues. Confirming earlier work in simulation, the robot was able to avoid about 90% of obstacles in a cluttered arena. In addition, the algorithm successfully guided the robot through a corridor lined with stochastic reflectors. Relying on interaural level differences, the Fixed Head Strategy outperformed the DLAL Strategy: locking the acoustic gaze to the driving direction increased performance.

## Acknowledgments

We thank Professor Claudio Rossi and an anonymous reviewer for comments on an earlier version of this manuscript.

## Author Contributions

**Conceptualization:** Carl Bou Mansour, Elijah Koreman, Herbert Peremans, Dieter Vanderelst.

**Formal analysis:** Dieter Vanderelst.

**Investigation:** Carl Bou Mansour, Elijah Koreman, Dieter Vanderelst.

**Methodology:** Elijah Koreman, Jan Steckel.

**Resources:** Jan Steckel.

**Software:** Jan Steckel.

**Supervision:** Dieter Vanderelst.

**Writing – original draft:** Herbert Peremans.

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
