## [Decision Letter · Decision Letter 0]

12 Sep 2019

Dear Dr Vanderelst,

Thank you very much for submitting your manuscript, 'Avoidance of non-localizable obstacles in echolocating bats: a robotic model', to PLOS Computational Biology. As with all papers submitted to the journal, yours was fully evaluated by the PLOS Computational Biology editorial team, and in this case, by independent peer reviewers. The reviewers appreciated the attention to an important topic but identified some aspects of the manuscript that should be improved.

We would therefore like to ask you to modify the manuscript according to the review recommendations before we can consider your manuscript for acceptance. Your revisions should address the specific points made by each reviewer and we encourage you to respond to particular issues Please note while forming your response, if your article is accepted, you may have the opportunity to make the peer review history publicly available. The record will include editor decision letters (with reviews) and your responses to reviewer comments. If eligible, we will contact you to opt in or out.raised.

- Supporting Information uploaded as separate files, titled 'Dataset', 'Figure', 'Table', 'Text', 'Protocol', 'Audio', or 'Video'.

We hope to receive your revised manuscript within the next 30 days. If you anticipate any delay in its return, we ask that you let us know the expected resubmission date by email at ploscompbiol@plos.org.

Sincerely,

Joseph Ayers, PhD

Associate Editor

PLOS Computational Biology

Wolfgang Einhäuser

Deputy Editor

PLOS Computational Biology

[LINK]

Reviewer's Responses to Questions

**Comments to the Authors:**

Reviewer #1: The paper applies recent findings in bats' obstacle-avoidance strategies to mobile robotics.

It is not clear if the purpose of the work is to provide robotics with novel obstacle-avoidance techniques, or if the robot platform is used to test biological hypothesis. I assume the latter, otherwise the lack of comparison with other obstacle avoidance techniques would make the paper quite weak.

So I have two major comments:

-The conclusions are based on a limited number of experiments (5 trials in the rectangular arena, 15 in the corridor) so their statistical significance is limited, I think that more experiments would make the paper much stronger.

-Regarding the comparison of fixed/scanning gaze, I think that this is a bit unfair since the poorer behaviour of the latter may be influenced by the speed of the scanning w.r.t. the robot's speed (boy linear and angular) and manoeuvring capabilities. So faster scanning and/or lower robot speed may produce less collisions (and vice-versa). This aspect should be considered and -if possible- assessed experimentally.

Minor comments:

-Please describe better which is the purpose of the random reflectors and of the egg cartoons in the corridor.

-Fig 1 is too small.

Reviewer #2: Referee’s comments on PCOMPBIOL-D-19-01284, Avoidance of non-localizable obstacles in echolocating bats: a robotic model.

Carl Bou Mansour, Elijah Koreman, Jan Steckel, Herbert Peremans, Dieter

Vanderelst

This is an excellent combination analysis of based on behavioral performance of echolocating bats and testing of a robotic model of obstacle avoidance that compares two different strategies for aiming the model bat’s head and sonar beam. In one model the beam is aimed straight ahead in the upcoming path, not side-to-side, so that the obstacles that surround the path are kept to the sides, leaving the upcoming path to be checked for whether it is open or obstructed. Behavioral data that support this method are described in a paper by Knowles, at al., 2015. In a cluttered scene involving choices about which potential upcoming path is unobstructed, the bat aims its beam into the path, turning slightly to anticipate and imminent steering action. It does not scan left and right. A paper (Temporal binding of neural responses for focused attention in biosonar. James A. Simmons Journal of Experimental Biology 2014 217: 2834-2843; doi: 10.1242/jeb.104380) examines the underlying perceptual mechanism. The alternative method, scanning left and right, is typical of bats flying in open, uncluttered spaces while searching for insects. This is well-documented in the manuscript’s references. One point worth mentioning is that the robotic model assumes that bats determine the horizontal direction of an object from the binaural difference in echo amplitude. The evidence suggests that the bat uses binaural echo delay differences and that the amplitude difference serves to magnify the time difference through the process of amplitude-latency trading, as described in the Pollak reference in the manuscript. These details need to be given in the manuscript because they strengthen the authors’ case.

**Have all data underlying the figures and results presented in the manuscript been provided?**

Reviewer #1: Yes

Reviewer #2: Yes

PLOS authors have the option to publish the peer review history of their article (what does this mean?). If published, this will include your full peer review and any attached files.

Reviewer #1: Yes: Claudio Rossi, Universidad Politécnica de Madrid

Reviewer #2: No

---

## [Decision Letter · Decision Letter 1]

17 Nov 2019

Dear Dr Vanderelst,

We are pleased to inform you that your manuscript 'Avoidance of non-localizable obstacles in echolocating bats: a robotic model' has been provisionally accepted for publication in PLOS Computational Biology.

In the meantime, please log into Editorial Manager at https://www.editorialmanager.com/pcompbiol/, click the "Update My Information" link at the top of the page, and update your user information to ensure an efficient production and billing process.

One of the goals of PLOS is to make science accessible to educators and the public. PLOS staff issue occasional press releases and make early versions of PLOS Computational Biology articles available to science writers and journalists. PLOS staff also collaborate with Communication and Public Information Offices and would be happy to work with the relevant people at your institution or funding agency. If your institution or funding agency is interested in promoting your findings, please ask them to coordinate their releases with PLOS (contact ploscompbiol@plos.org).

Thank you again for supporting Open Access publishing. We look forward to publishing your paper in PLOS Computational Biology.

Sincerely,

Joseph Ayers, PhD

Associate Editor

PLOS Computational Biology

Wolfgang Einhäuser

Deputy Editor

PLOS Computational Biology

Reviewer's Responses to Questions

**Comments to the Authors:**

Reviewer #1: The authors have addressed my previous comments and I have no further comments

Reviewer #2: good revision

**Have all data underlying the figures and results presented in the manuscript been provided?**

Reviewer #1: None

Reviewer #2: Yes

PLOS authors have the option to publish the peer review history of their article (what does this mean?). If published, this will include your full peer review and any attached files.

Reviewer #1: Yes: Claudio Rossi

Reviewer #2: No

---

## [Editor Report · Acceptance letter]

12 Dec 2019

PCOMPBIOL-D-19-01284R1 

Avoidance of non-localizable obstacles in echolocating bats: a robotic model

Dear Dr Vanderelst,

I am pleased to inform you that your manuscript has been formally accepted for publication in PLOS Computational Biology. Your manuscript is now with our production department and you will be notified of the publication date in due course.

With kind regards,

Matt Lyles
